# Intraosseous Bioplasty for a Subchondral Cyst in the Lateral Condyle of Femur

**DOI:** 10.3390/jcm9051358

**Published:** 2020-05-06

**Authors:** Anish G.R. Potty, Ashim Gupta, Hugo C. Rodriguez, Ian W. Stone, Nicola Maffulli

**Affiliations:** 1South Texas Orthopaedic Research Institute, Laredo, TX 78045, USA; anishpotty@gmail.com (A.G.R.P.); ashim6786@gmail.com (A.G.); hcrodrig@student.uiwtx.edu (H.C.R.); 2School of Osteopathic Medicine, University of the Incarnate Word, San Antonio, TX 78209, USA; istone@student.uiwtx.edu; 3Laredo Sports Medicine Clinic, Laredo, TX 78041, USA; 4Department of Psychology, Illinois Wesleyan University, Bloomington, IL 61701, USA; 5Future Biologics, Lawrenceville, GA 30043, USA; 6BioIntegrate, Lawrenceville, GA 30043, USA; 7Department of Musculoskeletal Disorders, School of Medicine and Surgery, University of Salerno, 84084 Fisciano, Italy; 8Barts and the London School of Medicine and Dentistry, Centre for Sports and Exercise Medicine, Queen Mary University of London, London E1 4DG, UK; 9School of Pharmacy and Bioengineering, Keele University Faculty of Medicine, Stoke on Trent ST4 7QB, UK

**Keywords:** intraosseous bioplasty, subchondral cysts, bone marrow edema, orthobiologics, autologous bone marrow, platelet rich plasma, demineralized bone

## Abstract

Several conditions can lead to the development of a subchondral cyst. The mechanism by which the cysts form, their location, and their severity depend on the underlying pathology, although the exact pathogenesis is not fully elucidated. Treatment options vary according to the location of the cyst, with less invasive procedures such as calcium phosphate cement injection to a joint arthroplasty when there is an extensive cyst in communication with the joint space. If the cyst is circumscribed, an intraosseous bioplasty (IOBP) can be performed. Described in this paper is an IOBP, a minimally invasive technique that preserves the joint and can be applied to most subchondral cysts. In our patient, both the appearance of the cyst at imaging and pain after IOBP greatly improved with the combined use of decompression and grafting. In those patients in whom conservative management fails to ameliorate symptoms, IOBP should be considered.

## 1. Introduction

Subchondral cysts (SC) are common in several conditions including osteoarthritis (OA), rheumatoid arthritis (RA), osteonecrosis, and calcium pyrophosphate deposition (CPD) disease [1,2,3,4]. These conditions lead to the development of bone marrow lesions, which are strong predictors of SC development [5]. These cysts are often benign and incidentally found on radiography, with 80% of patients reporting no symptoms unless a fracture occurs [6]. While the location of the cyst does not influence symptoms, knee pain is associated with the volume of the cystic component [7]. In OA of the hip, SC occur on the pressure segment of the femoral head along with the loss of articular space. In contrast, in RA, SC are initially noted at the osteo-chondral junction and may affect the entire femoral head. In osteonecrosis, the cysts develop in the necrotic segment of the femoral head, whereas in CPD disease, they resemble those in the OA but are larger, and more numerous and widespread [1,3].

The etiology of SC is still not well understood. Two theories—the “synovial intrusion theory” and the “bony contusion theory”—attempted to explain the formation of SC [1,8]. The former suggested that a SC could be secondary to the pathology of the synovium extending into the bone from the resemblance of synovial fluid to the cystic fluid, abnormal articular cartilage over the cyst, and dislocated pieces of hyaline cartilage within the cyst [9]. This is supported by the frequent communication between joint cavities and SC; however, this communication is not present in all SC. The “bony contusion theory” explained this absence of communication: the impact between opposing bone surfaces that have lost their protective cartilage can lead to microfractures and bone necrosis. The synovial fluid then intrudes the bone when this attempts to heal by osteoclastic resorption of the necrotic bone. This is supported by the lack of communication between the joint cavity and the subchondral cyst, the presence of the metaplastic cartilage, and the osteoclasts in the disrupted bone [10]. Another proposed mechanism of cyst formation is the overloading of specific areas within the bone from the unevenness of the articular surfaces [1,11]. This would increase the intraosseous pressure leading to decreased blood flow and impeding the body′s ability to heal itself [12].

Traditionally, SC have been managed using physical therapy with pharmacological agents such as non-steroidal anti-inflammatory drugs, corticosteroids, and viscosupplementation; intra-cyst calcium phosphate cement injection has also been performed [13,14]. When SC progress, and conservative management fails, surgical options in the knee including high tibial osteotomy, unicondylar knee arthroplasty, and total knee arthroplasty (TKA) can be considered [13,14]. These modalities have several limitations including increased risk of early mechanical complications and early revisions after TKA in young patients [15]. Intraosseous bioplasty (IOBP) is a viable alternative in these young patients in whom joint sparing is desirable. In addition, IOBP can be repeated to try and delay a TKA.

IOBP involves the structural decompression of the area of pathology, thus reducing intraosseous pressure, followed by injection of a mixture of concentrated platelet rich plasma (PRP) from autologous bone marrow and demineralized bone to allow bone remodeling to not only stabilize the lesion but to heal it.

We describe a patient in whom IOBP was utilized, outlining clinical pearls, and indications and benefits of IOBP.

## 2. Surgical Technique

### 2.1. Indications

IOBP is indicated in patients with imaging findings of a subchondral cyst. In these patients, the lesions have well defined borders, are easily identified at MRI, and patients report pain, stiffness, decreased range of motion and function. A patient′s age plays a key role in identifying the ideal candidates for this procedure. Patients between 30 and 55 with a high risk of a TKA in the near future are ideal candidates for IOBP. Patients should be compliant with the postoperative protocol of non-weightbearing for at least 3 weeks. In this case, the patient reported pain, stiffness and decreased range of motion and function. Along with the associated MRI findings, there was a high risk of subchondral bone collapse and possible TKA an IOBP.

The main contraindication for IOBP is a breach in the articular surface of the bone overlying the lesion. This would lead to extravasation of the PRP-DBM mixture and no physiologic remodeling of the bone, which would lead to further trauma to the joint. Patients with joint space narrowing, osteophytes, full thickness cartilage loss, and underlying bone exposure would prompt to consider other procedures rather than IOBP. Some relative contraindications if there is no articular cartilage breach could include other inflammatory, erosive, and crystal arthropathies. The size of the lesion is also important, as cysts > 2.5 cm in diameter have less optimal results (Table 1).

### 2.2. History and Diagnosis

The initial diagnosis begins with thorough history evaluation. In this case, the patient was a 47 years old male with a past medical history of chronic pain with no previous trauma to the right knee. A physical exam is also paramount, with emphasis on the knee. Stress testing of all the ligaments, with assessment of the knee range of motion and muscle strength, should be performed. This allows the clinician to assess for other intra-articular pathology, including ligament instability or meniscal trauma that might need to be addressed concurrently during the IOBP. A 6-foot standing alignment radiograph for varus or valgus malalignment should be performed, with a view to correction if necessary. Additionally, MRI evaluation allows precise localization of the lesion and planning of the procedure accordingly. MRI evaluation in the patient reported in this article evidenced a large multiloculated cystic area with internal septs in the lateral femoral condyle, close to the insertion of the popliteal tendon, measuring 2.3 × 2.1 × 2.0 cm (Figure 1A,B). A diagnostic arthroscopy was planned to evaluate the joint space and ensure this was not compromised.

### 2.3. Patient Setting

The patient was brought to the operating room and was placed supine followed by administration of general anesthesia. The right lower extremity was prepped and draped in the routine fashion. A fluoroscope with a sterile drape was used to localize the cyst. The position was maintained during the Intraosseous Bioplasty.

### 2.4. Diagnostic Arthroscopy

Standard antero-lateral and antero-medial portals were established, and a routine diagnostic arthroscopy was performed using a 4.0 mm 30° arthroscope. The aforementioned meniscal findings were confirmed and addressed; the articular surface of the femur was inspected to ensure that surface was not breached (Figure 2).

### 2.5. Bone Marrow Harvest and Preparation

The 4.0 mm 30° arthroscope was inserted through the antero-lateral portal that was used during the diagnostic arthroscopy and utilized for direct visualization of the intercondylar notch during the harvesting process. An 8G by 11 cm Jamshidi needle was introduced into the joint space through the antero-medial portal. It was impacted into the intercondylar notch using a hammer to the depth of 0.4 cm. Under the guidance of lateral knee fluoroscopy with 90° knee flexion to provide for precise anatomical location into the medullary canal, the arthroscopic fluid was then turned off and the inner stylus of the needle was removed. Two 30 mL syringes pretreated with anticoagulant citrate dextrose solution were then placed connected to the needle to harvest 60 mL of bone marrow aspirate. The bone marrow aspirate was slowly extracted while rotating and slowly withdrawing the needle to adequately fill the syringe (Figure 3). This location for bone marrow harvesting avoids the need to make an additional incision (traditionally this would be at the anterior superior iliac spine). The 60 mL of bone marrow was then placed into the Arthrex Angel CPRP and Bone Marrow Processing system (Arthrex, Inc., Naples, FL, USA) and processed as per manufacturer’s instructions. After centrifugation for a 17 min, the concentrated Platelet Rich Plasma (PRP) was mixed with 5 mL of demineralized bone matrix (DBM) until the consistency of a dense paste was reached. This mixture was ready to be introduced once the cyst had been decompressed. During the entire bone marrow harvest process, it is important to utilize serial fluoroscopy and arthroscopic visualization. This allows proper visualization of the location of the intercondylar harvest and ensures that there is no breech in the articular surface over the SC.

### 2.6. Core Decompression

While the Angel system was processing the 60 mL aspirate, the cyst was decompressed (Appendix A). With the aid of fluoroscopy and previous MRI images, the lesion at the lateral condyle of the femur was located using an 8G by 11 cm Jamshidi needle. A 1 cm long incision was made along the distal-lateral portion of the right femur over the lateral condyle under fluoroscopic guidance. The iliotibial band and proximal structures were retracted for adequate visualization of the lateral condyle of the femur. A 2.4 mm guide pin was then directly drilled into the lesion and used to ream it with a low profile 7 mm reamer (Appendix A). The arthroscope was introduced in the tunnel to visualize the wall of the cyst and assess whether further septae needed to be broken. It is important to ensure that the wall of the cyst is removed, and the underlying bone is actively bleeding. The loculations were then removed with the use of curettes, taking care that the opposite bone cortex was not compromised (Appendix A). This can be achieved by using serial fluoroscopy imaging as well as arthroscopy. The arthroscope was used to visually ensure that the articular surface of the femur was not breached.

### 2.7. PRP Inoculation and Bone Impaction

An Open-Tip 8G by11 cm delivery cannula was used (Figure 4A) to allow direct delivery into the lesion and maximum precision while the PRP-DBM mixture is being introduced. The delivery cannula was placed directly into the lesion under careful fluoroscopic guidance and arthroscopy to ensure that the opposite cortex was not penetrated and that the articular surface remained intact (Figure 5A). The mixture of 50:50 Isovue dye, demineralized bone matrix, and PRP in the 14 mL syringe was loaded on the delivery cannula (Figure 4A,B) and delivered under direct observation with fluoroscopy. It can be helpful to transfer the mixture into a smaller 1 mL syringe (Figure 4A) using an adapter that connects the two syringes if high resistance is present. The stylet from the cannula can also be used to push the mixture against resistance. Allograft cancellous bone was then impacted into the defect using a curette, and the remaining mixture was then injected while viewing the cyst with an arthroscope. Reinserting the stylet into the cannula at this time would make certain that all of the mixture is in the cyst. The arthroscope was then used to ensure that the injected material did not extravasate into the joint space. A final fluoroscopy assessment of the lesion was performed (Figure 5B).

### 2.8. Postoperative Protocol

Post-operatively, the patient was instructed to not to weight bear for 3 weeks, followed by 3 weeks of toe touch weight bearing, continuing with 3 weeks of partial weight bearing, and, finally, progress full weight bearing. The patient was seen 12 days following the procedure for suture removal. Serial Anteroposterior (AP) and Lateral radiographs were obtained at 3 and 6 months at follow up appointments to assess for adequate progressive filling of the defect and compared to pre-operative radiographs for reference. (Figure 6A–F). An Ossur CTI knee brace was used for all daily activities. The overall goal, that our patient has successfully achieved, was to have ROM to flex the knee to 0°–120°, have minimal or no pain with 50% activity, and gradually experience minimal pain 80% of the day.

## 3. Discussion

IOBP is a viable option to manage SC in selected patients avoiding an open approach [16]. Calcium phosphate and cement have been used, but they are brittle, and their use is usually limited to non-load-bearing areas [17]. In IOBP, the combination of decompression and autologous bone marrow grafting allows achievement of a long-lasting solution as well as naturally promoting regrowth of the affected bone [18,19,20,21,22]. In this way, the structural integrity of the joint space is maintained, the tissue where the cyst has developed can be restored [19], as shown in the postoperative imaging and improved postoperative symptoms [21]. Ultimately, this procedure allows to spare the joint and delay the need for major procedures and is particularly indicated in younger individuals. IOBP carries minimal risks for adverse events, as it employs autologous bone marrow graft (Table 2).

Careful assessment of the cyst and joint cavity should be performed both pre- and intra-operatively, as communication between the two would lead to extravasation of the graft material, and it is a definite contraindication to IOBP.

It is imperative to ensure adequate pin placement through fluoroscopy prior to decompressing the cyst. Delivery of the biologic material should be performed under direct visualization under fluoroscopy, as a radio-opaque dye is added to the biologic mixture to ensure proper location of the injection and direct delivery of the PRP-DBM into the lesion.

The cancellous bone graft impacted into the defect contains high concentrations of osteoblasts and osteocytes, conferring a high osteogenic capacity. Its large trabecular surface area confers stability and encourages revascularization (Table 3). A local hematoma forms: it is rich in inflammatory cells and chemotactic mitogens that recruit mesenchymal stem cells to the defect, leading to neovascularization, osteoid deposition, and ultimately mineralization leading to new bone [22].

In conclusion, IOBP is a minimally invasive procedure which provides a permanent and biologic solution for SC. IOBP gives clinicians a versatile way of utilizing the physiological principles involved in bone remodeling to not only stabilize the lesion but to heal it. This is in accordance with a recently published article where IOBP was used to treat in similar lesion in lateral Tibial plateau [23]. With successful treatment, the pain can be addressed while ensuring that the joint integrity is preserved.

## Figures and Tables

**Figure 1 jcm-09-01358-f001:**
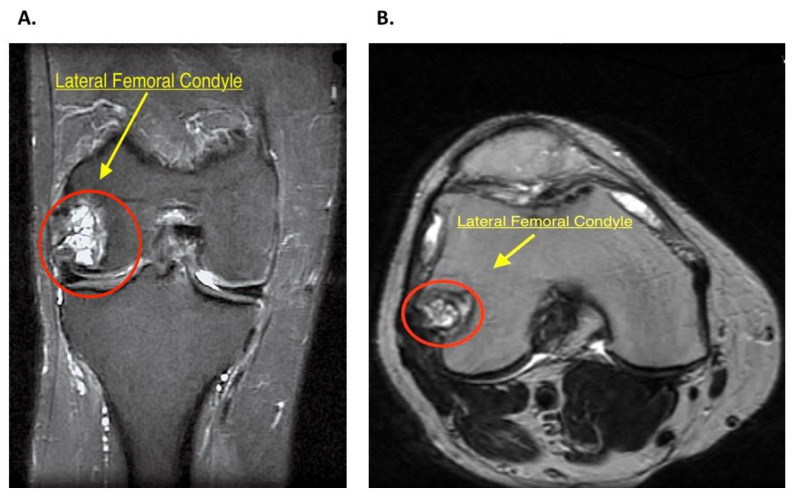
(**A**) Right knee magnetic resonance image, fat-saturated FSE-IR (fast spin-echo inversion-recovery). Coronal plane shows a multiloculated cystic area (red circle) in the lateral femoral condyle with proximal bone marrow edema. (**B**) Right knee magnetic resonance image, non-fat-saturated T2 weighted. Axial plane image shows the cystic area (red circle) used for localization and planning of procedure.

**Figure 2 jcm-09-01358-f002:**
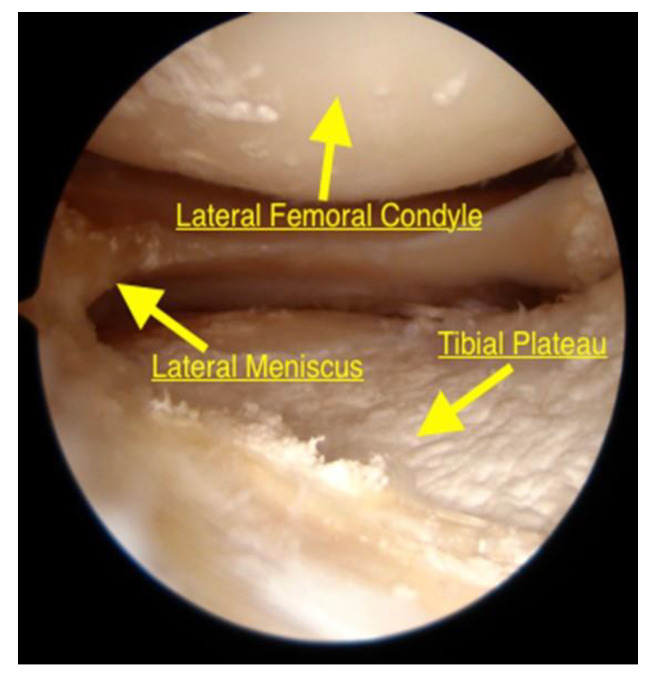
Arthroscopy appearance of the lateral compartment of the knee from the antero-lateral portal. The knee is in the figure-four position to allow adequate visualization of the lateral joint space. The lateral condyle was intact. Crystal deposits are evident in the articular cartilage.

**Figure 3 jcm-09-01358-f003:**
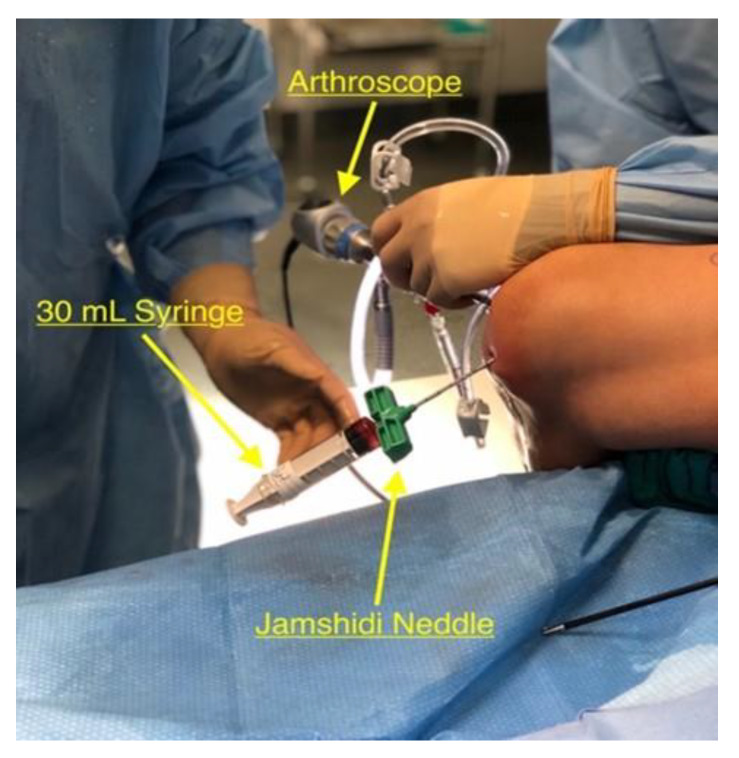
Harvest setup. The knee is flexed over the side of the table and the Jamshidi needle is impacted into the intercondylar notch of the femur with a pretreated 30 mL syringe. The arthroscope is in the anterior-lateral portal for visualization.

**Figure 4 jcm-09-01358-f004:**
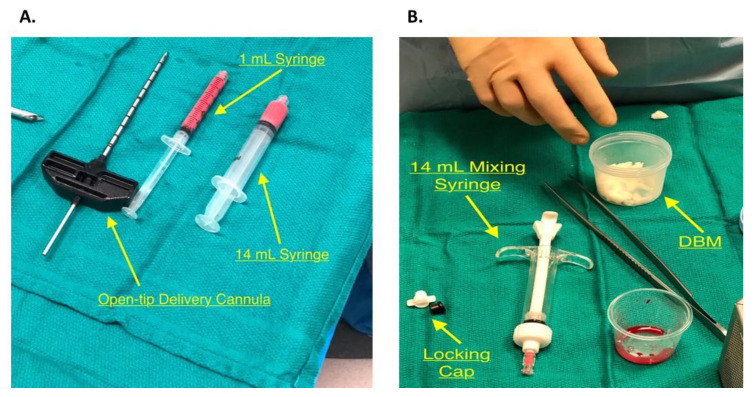
(**A**) The open-tip cannula is used for the direct approach of Intraosseous Bioplasty. Removal of the stylet will show the circular open end, rather than the 3-pin side end delivery used in the indirect approach. 1 mL syringe and 14 mL syringe filled with 50:50 Isovue dye, Demineralized Bone Matrix (DBM) and Platelet Rich Plasms (PRP). (**B**) The DBM in the plastic container will be mixed with the PRP from the Angle system in the container and will be inserted into the 1 mL or 14 mL syringes seen in Figure 4A.

**Figure 5 jcm-09-01358-f005:**
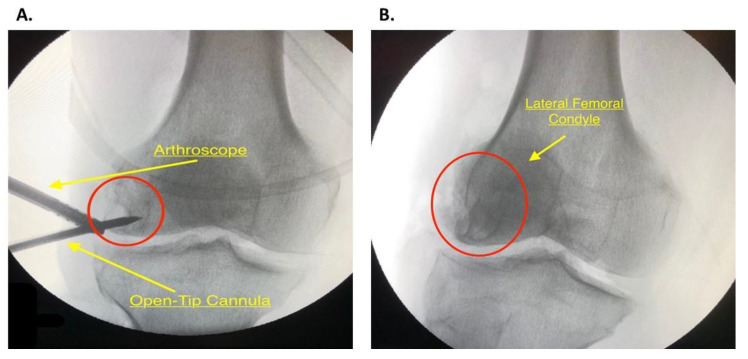
(**A**) Fluoroscopic image of the open-tip delivery cannula injecting the biologic mixture into the decompressed Subchondral Cyst (SC) (red circle) with arthroscopic guidance. (**B**) Final fluoroscopic image of the right knee after injection of the biologic mixture into the SC (red circle).

**Figure 6 jcm-09-01358-f006:**
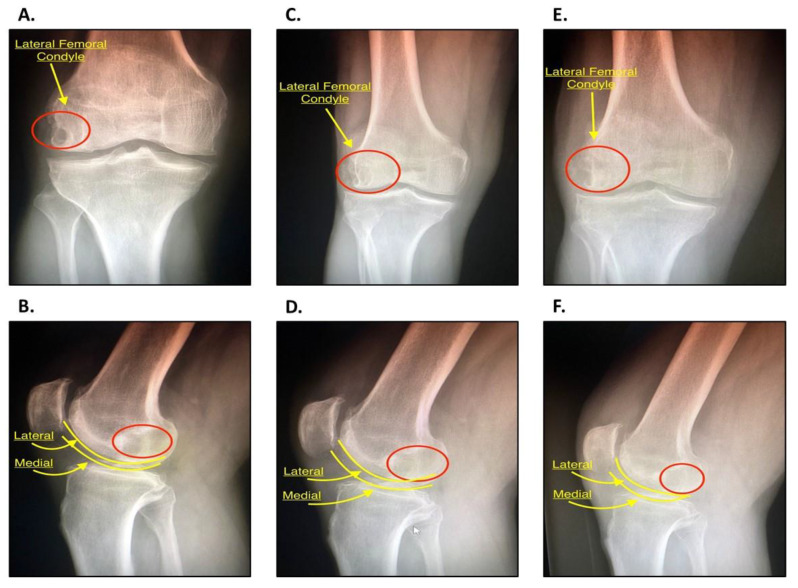
(**A**) Anteroposterior (AP) radiograph of the right knee, pre-operative. The lateral femoral condyle can be visualized with the area of decreased opacity representing the SC (red circle). The joint space is well preserved with mild patellofemoral OA with osteophytes. (**B**) Lateral radiograph of the right knee, pre-operative. The red circle depicts the same SC. The lateral and medial femoral condyles are outlined in yellow. Both (**A**) and (**B**) will be used as a baseline for comparison. (**C**) AP radiograph of the right knee 3 months following IOBP. The lateral femoral condyle can be visualized with signs that the area of the previous SC is filling (red circle). (**D**) Lateral radiograph of right knee 3 months following operation. The red circle depicts the area of the SC. (**E**) AP radiograph of the right knee 6 months following IOBP. The area of the previous SC (red circle) has increased in opacity in comparison to the previous image at 3 months. (**F**) Lateral radiograph of the right knee six months following IOBP. The area of the previous SC (red circle) is more opaque, an indication that there is progressing filling of the previous lesion. This suggest that the IOBP has been successful.

**Table 1 jcm-09-01358-t001:** Indications, Contraindications and Relative Contraindications of Intraosseous Bioplasty.

Indications	Contraindications	Relative Contraindications
A subchondral cyst with well-defined borders Symptoms such as pain, stiffness and decreased range of motion and function Patient aged between 30 and 55 Avoidance of Total Knee Arthroplasty	Breach in the articular surface of overlying bone Evidence of Grade 4 osteoarthritis	Inflammatory, erosive and crystal arthropathies Lesion size > 2.5 cm

**Table 2 jcm-09-01358-t002:** Advantages and Disadvantages of IOBP.

Advantages	Disadvantages
Minimally invasive Joint preservation, avoiding arthroplasty Wide choice of revision options Biological solution: physiological remodeling used for stabilization	Intraoperative time needed to prepare PRP-DBM solution Bone marrow harvest is needed, thus might need to have an extra incision site Unable to perform if overlying bone is breached

**Table 3 jcm-09-01358-t003:** Pearls and Pitfalls of IOBP.

Pearls	Pitfalls
Directly visualize the articular surface via arthroscopy to asses articular surface Harvest bone marrow without making an additional incision Perform the core decompression during the centrifugation Use a guide pin during the decompression process Perform a tunnel scope to for direct visualization Impact allograft cancellous bone into the decompressed lesion to add structural stability and increase revascularization Use a 1 mL syringe or utilize the inner stylet during PRP Inoculation	Failing to do serial fluoroscopy and arthroscopy Breach in the opposite cortex or articular surface during decompression

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
