# Peer review of "Intraosseous Bioplasty for a Subchondral Cyst in the Lateral Condyle of Femur"

_jcm, 2020, doi:10.3390/jcm9051358_

Round 1

Reviewer 1 Report

The author described a method using a minimally invasive technique intraosseous bioplasty (IOBP) to remove subchondral cysts and fill with autologous PRP-DBM mixture. The author provided very clear pre-surgical indications and contraindications to be considered for IOBP and described the diagnostical imaging of the patient case.

Overall, the surgical procedure is well designed with a clear advantage over the traditional methods such as TKA, to allow the autologous PRP graft within the same surgery to promote the regrowth of the affected bone. The harvest of bone marrow from intercondylar notch avoids additional incision and save procedure time. The introduction of allograft cancellous bone graft to the mix further enhances the osteogenic capacity. The author explained the surgical procedure with sufficient details. It would be great if the author would explain the usage of fluoroscopic and arthroscopic guidance in the bone marrow harvest and PRP-DBM mixture in more detail.

Author Response

Date: April 30, 2020

The Editor and Reviewer(s),

Journal of Clinical Medicine

Dear Sir/Madam,

We would like to thank you and the reviewers for reviewing our manuscript. We appreciate your comments and suggestions to improve the quality of our manuscript.  

We appreciate the opportunity to resubmit a revised manuscript after addressing the reviewers’ comments. Below you will find the response to each of the reviewer’s specific comments with the corresponding changes that are made throughout the manuscript. The changes made in the text in the manuscript are highlighted in bold and underlined.

Thank you in advance for your consideration. We look forward to hearing your editorial decision and, should they be necessary, welcome any further suggestions.

Sincerely,

Nicola Maffulli, MD, MS, PhD, FRCP, FRCS (Orth)

Professor of Trauma and Orthopaedic Surgery

Consultant Trauma and Orthopaedic Surgeon

REPLY TO REVIEWERS

Reviewer 1 Comments:

  1. The author described a method using a minimally invasive technique intraosseous bioplasty (IOBP) to remove subchondral cysts and fill with autologous PRP-DBM mixture. The author provided very clear pre-surgical indications and contraindications to be considered for IOBP and described the diagnostical imaging of the patient case.

Overall, the surgical procedure is well designed with a clear advantage over the traditional methods such as TKA, to allow the autologous PRP graft within the same surgery to promote the regrowth of the affected bone. The harvest of bone marrow from intercondylar notch avoids additional incision and save procedure time. The introduction of allograft cancellous bone graft to the mix further enhances the osteogenic capacity. The author explained the surgical procedure with sufficient details. It would be great if the author would explain the usage of fluoroscopic and arthroscopic guidance in the bone marrow harvest and PRP-DBM mixture in more detail.

Thank you for the comment. The details related to usage of fluoroscopic and arthroscopic guidance in the bone marrow harvest and PRP-DBM mixture has been added in Lines 145-151 & 161-163 The following text has been added:

  • “The 4.0 mm 300 arthroscope was inserted through the antero-lateral portal that was used during the diagnostic arthroscopy and utilized for direct visualization of the intercondylar notch during the harvesting process. An 8G by 11 cm Jamshidi needle was introduced into the joint space through the antero-medial portal. It was impacted into the intercondylar notch using a hammer to the depth of 0.4 cm. Under the guidance of lateral knee fluoroscopy with 900 knee flexion to provide for precise anatomical location into the medullary canal, the arthroscopic fluid was then turned off and the inner stylus of the needle was removed.” (Lines 145-151)
  • “During the entire bone marrow harvest process, it is important to utilize serial fluoroscopy and arthroscopic visualization. This allows proper visualization of the location of the intercondylar harvest and ensures that there is no breech in the articular surface over the SC. . “ (Lines 161-163)

Reviewer 2 Comments:

  1. This is a nice brief report depicting a successful IOBP of a subchondral cyst in the lateral femoral condyle of single patient. However, the basic subject information – gender, age, follow-up symptoms – is missing, and should be added to further elucidate this case.

Thank you for the comment. The following has been added into the text in Lines 103-105 & 230-232:

  • “The initial diagnosis begins with thorough history evaluation. In this case, the patient was a 47 year old male with a past medical history of chronic pain with no previous trauma to the right knee. A physical exam is also paramount…” (Lines 103-105)
  • “The overall goal, that our patient has successfully achieved, was to have ROM to flex the knee to 0-120°, have minimal or no pain with 50% activity, and gradually experience minimal pain 80% of the day. (Lines 230-232)
  1. Another thing bothering me is the suggested association between the subchondral cyst and symptoms, which is highly arguable; I think no causality actually exists, but disease underlying the cyst formation causes the symptoms (e.g. pain, joint stiffness) in general. The main indication for cyst treatment should be the prevention of imminent fracture or progression of the underlying disease.

This suggested comment has been addressed and the following has been added in lines 42-44 & 88-90

  • “These cysts are often benign and incidentally found on radiography, with 80% of patients reporting no symptoms unless a fracture occurs [6]. While the location of the cyst does not influence symptoms, knee pain is associated with the volume of the cystic component [7].” (Lines 42-44)
  • “In this case, the patient reported pain, stiffness and decreased range of motion and function. Along with the associated MRI findings, there was a high risk of subchondral bone collapse and possible TKA an IOBP.”(Lines 88-90)

The more specific minor comments are shown below:

  1. Abstract:

Line 29: There´s typo here: should read joint not join.

The correction has been made.

  1. Line 31: This sentence is hard to read, please re-phrase.

The suggested change has been made. The sentence has been re-phrased to (Lines 30-32):

  • Described in this paper is an IOBP, a minimally invasive technique that preserves the joint and can be applied to most subchondral cysts.
  1. Introduction:

How many of the SCs are symptomatic, is there any literature on that? And obviously the size and specific locations of the SCs increase the risk for complication (i.e. fracture), but this should also read in the Introduction, since the majority of the SCs are non-significant clinically.

The suggested change has been made. The following has been added to the text in Lines 42-44:

  • These cysts are often benign and incidentally found on radiography, with 80% of patients reporting no symptoms unless a fracture occurs [6]. While the location of the cyst does not influence symptoms, knee pain is associated with the volume of the cystic component [7].
  1. Surgical technique:

Line 80: Is there actually any association between the SC and symptoms? I dare to argue that in most cases virtually no, but rather the underlying OA/RA is the entity causing symptoms. I think IOBP´s main indication should be to avoid possible fracture due to the SC.

This suggested comment has been addressed and the following has been added in lines 42-44 & 88-90:

  • “These cysts are often benign and incidentally found on radiography, with 80% of patients reporting no symptoms unless a fracture occurs [6]. While the location of the cyst does not influence symptoms, knee pain is associated with the volume of the cystic component [7]. (Lines 42-44)
  • “In this case the patient reported pain, stiffness and decreased range of motion and function. Along with the associated MRI finding, high risk of subchondral bone collapse and possible TKA an IOBP was indicated.”(Lines 88-90)
  1. Figure 1:

Line 108: Typo here, should read plane not plain.

The correction has been made.

Both MR images are non-fat-saturated, which should read in the legend.

The correction has been made.

Moreover, axial plane really isn´t good for assessing the close relation of the SC to the joint space, so please remove this comment.

The correction has been made. And the following text has been added in lines 121-122:

  • .Axial plane image shows the cystic area (red circle) used for localization and planning of procedure. “
  1. T1 or PD/T2 weighted fat-saturated sequences could also provide more information on the SC for this figure; for instance, is there any bone marrow edema present adjacent to the SC.

We agree with the reviewer. We have switched the previous non-fat saturated MRI to an FSE-IR Coronal MRI. The new FSE-IR Coronal MRI image and caption shown below:

The Figure caption now reads: “ Figure 1. (A) Right Knee magnetic resonance image, fat-saturated FSE-IR weighted. Coronal plane shows a multiloculated cystic area (red circle) in the lateral femoral condyle with proximal bone marrow edema. (B) Right Knee magnetic resonance image, non-fat-saturated T2 weighted. Axial plane image shows the cystic area (red circle) used for localization and planning of procedure. “

  1. Core decompression:

The text should imply that a lateral extra-articular route was used for the procedure.

The suggested change has been made. The following has been added to the text in Lines 177-181:

  • “A 1 cm long incision was made along the distal-lateral portion of the right femur over the lateral condyle under fluoroscopic guidance. The iliotibial band and proximal structures were retracted for adequate visualization of the lateral condyle of the femur. A 2.4 mm guide pin was then directly drilled into the lesion and used to ream it with a low profile 7 mm reamer (Video S1).
  1. Figure 6 & 7:

The pre-operative radiography should be also introduced, and all the radiographs shown in 3 x 2 figure to allow easy comparison. Of note, the obvious OA changes should be commented, as well as the gender and age of the patient. Post-operative MRI could be intriguing, but most likely it was not performed?

We agree with the reviewers’ comment. We have added a 3 x 2 radiograph image that includes the pre-operative AP and lateral images as well as 3 and 6 months follow up images.

The following text had been added in lines (227-229):

  • “Serial Anteroposterior (AP) and Lateral radiographs were obtained at 3 and 6 months at follow up appointments to assess for adequate progressive filling of the defect and compared to pre-operative radiographs for reference. (Figures 6A-F).”

The following Figure caption has been added in lines 237-249. Comments on OA changes are addressed as well:

  • Figure 6. (A) AP radiograph of the right knee, pre-operative. The lateral femoral condyle can be visualized with the area of decreased opacity representing the SC (red circle). The joint space is well preserved with mild patellofemoral OA with osteophytes. (B) Lateral radiograph of the right knee, pre-operative. The red circle depicts the same SC. The lateral and medial femoral condyles are outlined in yellow. Both 6A and 6B will be used as a baseline for comparison. (C) AP radiograph of the right knee 3 months following IOBP. The lateral femoral condyle can be visualized with signs that the area of the previous SC is filling (red circle). ). (D) Lateral radiograph of right knee 3 months following operation. The red circle depicts the area of the SC. (E) AP radiograph of the right knee 6 months following IOBP. The area of the previous SC (red circle) has increased in opacity in comparison to the previous image at 3 months. (F) Lateral radiograph of the right knee six months following IOBP. The area of the previous SC (red circle) is more opaque, an indication that there is progressing filling of the previous lesion. This suggest that the IOBP has been successful.”

Comments on patients age and gender are addressed in lines 103-105:

  • “The initial diagnosis begins with thorough history evaluation. In this case, the patient was a 47 year old male with a past medical history of chronic pain with no previous trauma to the right knee. A physical exam is also paramount…”

We do agree with the reviewer that a Post-op MRI would be great due to the greater amount of detail. However, it was not done in this case and it is usually not indicated unless problems post-operatively occur.

  1. Discussion:

Line 216: For consistency, please use the abbreviation SC for the subchondral cyst.

The abbreviation SC for subchondral cyst is now consistent throughout the paper.     

  1. Line 270 Discussion could also benefit from adding the following article to it: Elena N, Woodall BM, Lee K, McGahan PJ, Pathare NP, Shin EC, Chen JL. Intraosseous Bioplasty for a Chondral Cyst in the Lateral Tibial Plateau. Arthrosc Tech. 2018 Oct 15;7(11):e1149-e1156. doi: 10.1016/j.eats.2018.07.011

The authors agree with the reviewer. The reference has been added and cited in the discussion (Reference number 23). The following statement has been added in Lines 303-304:

  • This is in accordance with a recently published article where IOBP was used to treat in similar lesion in lateral Tibial plateau.[23]

We thank the Reviewers for the precious input, which have allowed us to greatly improve the manuscript. We hope that the standard of our contribution has improved, and it has now reached the threshold necessary for acceptance in the Journal of Clinical Medicine.

We look forward to hearing from you.

Best regards

Nicola Maffulli

Reviewer 2 Report

This is a nice brief report depicting a successful IOBP of a subchondral cyst in the lateral femoral condyle of single patient. However, the basic subject information – gender, age, follow-up symptoms – is missing, and should be added to further elucidate this case. Another thing bothering me is the suggested association between the subchondral cyst and symptoms, which is highly arguable; I think no causality actually exists, but disease underlying the cyst formation causes the symptoms (eg. pain, joint stifness) in general. The main indication for cyst treatment should be the prevention of imminent fracture or progression of the underlying disease. The more specific minor comments are shown below:

Abstract:

Line 29: There´s typo here: should read joint not join.

Line 31: This sentence is hard to read, please re-phrase.

Introduction:

How many of the SCs are symptomatic, is there any literature on that? And obviously the size and specific locations of the SCs increase the risk for complication (i.e. fracture), but this should also read in the Introduction, since the majority of the SCs are non-significant clinically.

Surgical technique:

Line 80: Is there actually any association between the SC and symptoms? I dare to argue that in most cases virtually no, but rather the underlying OA/RA is the entity causing symptoms. I think IOBP´s main indication should be to avoid possible fracture due to the SC.

Figure 1:

Line 108: Typo here, should read plane not plain.

Both MR images are non-fat-saturated, which should read in the legend. Moreover, axial plane really isn´t good for assessing the close relation of the SC to the joint space, so please remove this comment. T1 or PD/T2 weighted fat-saturated sequences could also provide more information on the SC for this figure; for instance, is there any bone marrow edema present adjacent to the SC.

Core decompression:

The text should imply that a lateral extra-articular route was used for the procedure.

Figure 6 & 7:

The pre-operative radiography should be also introduced, and all the radiographs shown in 3 x 2 figure to allow easy comparison. Of note, the obvious OA changes should be commented, as well as the gender and age of the patient. Post-operative MRI could be intriguing, but most likely it was not performed?

Discussion:

Line 216: For consistency, please use the abbreviation SC for the subchondral cyst.

Discussion could also benefit from adding the following article to it: Elena N, Woodall BM, Lee K, McGahan PJ, Pathare NP, Shin EC, Chen JL. Intraosseous Bioplasty for a Chondral Cyst in the Lateral Tibial Plateau. Arthrosc Tech. 2018 Oct 15;7(11):e1149-e1156. doi: 10.1016/j.eats.2018.07.011

Author Response

(The authors gave the same response as above.)
